# Disability in Older People and Socio-Economic Deprivation in Italy: Effects on the Care Burden and System Resources

Georgia Casanova [1,2] and Roberto Lillini [3,*]

1 Instituto de Investigación en Políticas de Bienestar Social (POLIBIENESTAR)—Research Institute on Social Welfare Policy, Universidat de València, 46022 Valencia, Spain; georgia.casanova@uv.es

2 Centre for Socio-Economic Research on Ageing, IRCCS-INRCA National Institute of Health & Science on Ageing, 60124 Ancona, Italy

3 Analytical Epidemiology and Health Impact Unit, Fondazione IRCCS Istituto Nazionale dei Tumori, 20133 Milan, Italy

* Correspondence: roberto.lillini@istitutotumori.mi.it; Tel.: +39-022-390-2867 or +39-347-303-9225

**Abstract:** The sustainability of European Long-Term Care systems faces the demographic and socio-economic circumstances, mainly the increasing ageing of the population, with its chronic disease conditions, and the simultaneous economic general crises, exacerbated by the recent COVID-19 pandemic. Beyond the increase in general rate of relative poverty, there is a higher risk of poverty among elderly and families in a high demand of care, especially if situations of Activities Daily Living (ADL) disability are present. Italian welfare, which is based on family care regimes and regional strategies, and is oriented to private or public care, is a relevant case study with which to analyze such a relationship. This paper aims to study the relationship between ADL disability and the socio-economic deprivation of families, that is, household poverty. Variables came from the ISTAT Health for All Italian Database and the INAIL Disability Allowance Database. A pool of statistical methods, based on bivariate and multivariate analyses, from bivariate correlation, through multiple linear regression to principal component factor analysis, were used to reduce the number of the variables and compute the indicators. The multivariate analysis underlines how ADL disability impacts on a household's poverty, confirming the existence of statistical correlation between them. Moreover, the study identifies and measures two answer capability models to cope with household poverty. The answer capability of the formal system is the main tool for reducing poverty due to one family member's ADL disability. Integration and collaboration between the formal system and family capabilities remains the main solution.

**Keywords:** socio-economic deprivation; ADL disability; ageing; poverty; indicators

## 1. Introduction

Activities of Daily Living (ADL) disability, ageing, and poverty are central themes in the social and health policy debate. In particular, Long-Term Care (LTC) has been the subject of many interventions and reform proposals across Europe [1–6]. In 2017, Mosca and colleagues underlined how the sustainability of European LTC systems seemed to be fragile due to the demographic and fiscal circumstances and the complexities of LTC systems [7]. Furthermore, these authors emphasised the usefulness of learning from the policy design and implementation of LTC policy in other countries, within and outside of the EU. The literature identifies the informal care provided by relatives as one of the LTC pillars in many European countries [8,9]. In recent decades, low income and poverty risk issues have been relevant causes of social exclusion for older people [10,11]. Absolute poverty is a condition where the household income is below a necessary level to maintain basic living standards (food, shelter, housing). Janković-Milić et al. [10] underlined that most European countries use the baseline of relative income poverty. Persons below 60% of the median income in the EU are considered to be at risk of poverty. The existing European economic

crisis—which started in 2008—has focused the attention of public opinion and political decisionmakers towards the problem of low income and the capability of purchasing care services [12,13]. Recent studies have stressed how the poverty risk in old age might also be affected by welfare support provided by governments and other policy strategies used in different countries [14,15]. Therefore, the study and assessment of the relationship between LTC, ADL disability, and socio-economic deprivation is crucial [16].

Italy is the oldest European country and one of the lowest public investors in measures for social inclusion (1.7% of GDP). Moreover, its public intervention is ineffective in diminishing the risk of relative poverty [17,18]. For these reasons, Italy is a relevant case study on these topics in the international policy debate.

In 2018, almost 22.6% of the population was aged 65+, and more than half of older people were aged 75+ [12]. Simultaneously, relative poverty in Italy affects almost 12% of the population (more than 9 million people), and 8.4% Italians were in conditions of absolute and relative poverty (more than 1.8 million families). In comparison to 2012, when more than 3 million families were in absolute poverty—achieving the highest values ever recorded since the end of World War II—, the trends are slowly decreasing, though are always one of the highest in Europe [19,20]. Considering the 65+ population, in 2018, a 4.6% of persons in absolute poverty was recorded (corresponding to almost 630,000 persons), increasing to 4.8% in 2019 [20]. Thus, in contrast to the general trend, the Italian care regime is characterized by a familial care model, in which a very high level of LTC demand is traditionally met by informal/family care, and only to a minor extent by public in-kind services [21–23]. Therefore, an increase in the family burden due to LTC can easily be forecast [24,25]. Moreover, the Italian care system's high fragmentation underlines many differences at the local level, region by region. The Italian local LTC systems meet characteristics of the existing European LTC system, increasing the relevance of Italy as a European case study [26,27]. In some northern regions (e.g., Lombardy, Veneto, and Friuli-Venezia Giulia), the role of the private care provider is relevant, while a public strategy is being pursued in other northern and central regions (e.g., Tuscany, Emilia-Romagna, Trentino-Alto Adige and Valle d'Aosta). Otherwise, some regions promote the involvement of NGOs supporting the public sector (e.g., Liguria, Puglia and Piedmont), while families maintain a stronger role in southern regions (e.g., Sicily, Campania, Calabria) [26,28,29].

All these aspects of the problem can be contextualized in the following conceptual framework. The existence of a link between serious ADL disability and the condition of socio-economic deprivation is more than a simple hypothesis. Beyond the increase in the general rate of relative poverty, there is a higher risk of poverty among the elderly and families in high demand of care [30–32].

Thus, the strategies used to cover care needs are the object of this study, and include, in particular:

- Public strategy based on formal public services (national and local health and social systems, municipalities, etc.) [5,33];
- Private strategy based on the private care market, including care workers [34,35];
- The family-based strategy, based on informal care-giving [36,37].

The three strategies, generally used in an integrated rather than an exclusive manner, have an economic impact, which is direct if the services are bought, or indirect, if the family's economic contribution is determined by a reduction in the possibilities of financial entries.

In such a context, the combined effects of ageing and ADL disability, which inherently involve specific LTC needs, are relevant factors in increasing families' risk of poverty and are driving public health and social systems to find both preventive and supportive strategies and actions.

This study aims to define the correlation between ADL disability and the increasing poverty of families in Italy and to evaluate the effects on families, describing the main accountable factors.

Moreover, data covered all the national territory but were collected at the regional level, in order to give a better description of the role of the local differences in a de-centralized system.

These results should provide useful information to be discussed and used at the Italian and European level as a contribution toward improving policies.

## 2. Materials and Methods

Variables were taken from the official sources of the national statistical administrations, more specifically, the Italian National Statistics Office (ISTAT) and the Italian National Disability Allowance Office (INAIL), collected in the ISTAT Health for All Database [38].

The dependent variable was the Incidence of Household Poverty (Household Poverty, HP), expressed as the % ratio between the number of poor families and the number of families in the Health for All database at regional level. All the other variables from the four databases were the covariates.

Working on such a high number of variables compelled us to apply different steps of analysis in order to arrive at a simplification of the correlation structure. These were examined without losing calculative power. Four consecutive steps were performed in order to arrive at the final results.

(1) Analytic evaluation of the basic variables. The basic variables came from different periods (contained in the interval 1980–2015). Therefore, a preliminary standardization was applied in order to allow their comparability and their combination to be used in more complex models. In this way, it was possible to appreciate how they all showed a sufficiently regular distribution for each Italian region, without the presence of outliers that could compromise their use. Each variable's process of standardization used a historical trend of standard deviation as the normalizing element of the weighted average, considering the values observed year by year. In the next steps of the analysis, such standardized mean values (representing the total of the variable trend) were used [39].

(2) Bivariate comparisons. An initial aim of the analysis was to evaluate which variables could be used as indicators of poverty and which variables, when inserted in a multivariate model, could work as covariates compared to the indicators of poverty.

To reach this goal, a series of comparisons were made between all the possible pairs of variables through the Pearson correlation coefficient (statistical significance threshold at $p \leq 0.05$) (data not shown).

In this way, the following variables were identified:

- HP as an indicator of fundamental poverty [40]. All other associated variables correlated to this indicator with statistically significant values between 0.78 and 1.00 (as was expected, a superimposition was observed which was practically total with the incidence of individual poverty and the socio-economic deprivation index). Since families were the subject of the study, the HP indicator was chosen as the regional descriptor of poverty. Those variables which would have generated collinearity problems were discarded from the multivariate analysis. Moreover, the strong collinearity between HP (mainly, an indicator of material deprivation) and the socio-economic deprivation index (which also considers social aspects linked to social support, personal networks, etc.) also enabled the consideration of family needs related to economic aspects;
- Variables which outlined a significant correlation with indicators of poverty, if taken individually. To value the utility of the former within the multivariate models, a tolerance test (threshold value $p \leq 0.001$) [41] was performed to also analyse whether the correlations between the said variables led to situations of collinearity;

This process allowed:

- A reduction in the number of variables to 27 variables plus the HP;
- The identification of a series of information fundamental for the initial interpretation of family poverty conditions and for the definition of a simplified group of variables.

(3) Models of multivariate analysis. In order to evaluate the combined role of the covariates on the effects of family poverty, models of multiple linear regression were used (backward stepwise) with a control of partial collinearity in order to avoid distortions linked to the presence of eventual spurious relationships [42]. In this way, two models were computed which took into consideration, respectively, the combined effects of disability and the health system's capacity of response (independent covariates) on the incidence of poverty (dependent variable) and the combined role of disability, familial composition, employment, and unemployment (independent covariates) on the incidence of poverty (dependent variable). The models' goodness-of-fit was evaluated with adj. $R^2$ (statistical significance at $p \leq 0.05$) [41].

(4) Summarizing indicators and regional distribution. In order to provide a synthetic indicator at regional level, an index for the response capacity of services linked to disability and an index for the role of familial dimensions linked to employment/unemployment and disability were computed by a simple factor analysis (1 factor extracted per analysis). Both indices were rescaled from 0 (= lowest capability answer) to 10 (= highest capability answer) [41].

The scatter-plots, which put together the combination of these indices and the incidence of poverty in families at regional level, provided information regarding the role of the public offer and of the familial resources used to reduce the increasing effect of disabilities on poverty.

Record linkage and analyses in both steps of the study were performed with statistical software IBM SPSS 19.0 (IBM Corp. Armonk, NY, USA) and Stata 14.0 (StataCorp., College Station, TX, USA).

## 3. Results

Table 1 reports how every selected variable correlated to the incidence of HP. Large families, low education level, and unemployment, along with the presence of ADL disability, elderly people to be cared for at home, and regional out-of-pocket contribution, each stimulated an increase in HP.

**Table 1.** Variables correlated with the incidence of household poverty (HP) (statistical significance at $p < 0.05$).

| HP Increasing at Independent Variables Increase | HP Decreasing at Independent Variables Increase |
| --- | --- |
| Standardized ADL Limitations Rate, 6+ years old, M + F | Average amount of total disability pensions |
| Current public health expenditure in total agreement for social benefits (%) | % 1 person families |
| % 4 persons families | % 2 persons families |
| % 5 persons families | % 3 persons families |
| % 6+ persons families | Rate of healthy people, 65+ years old |
| % people with no education or primary level | % Local Public Health Units with integrated home care service |
| Unemployment rate, 15+ years old | Cases treated by integrated home care |
| % of elderly people (65+) treated in home care | Integrated home care rate |
| Index of territorial coverage of home care for the elderly (65+) | Integrated home care rate, 65+ years old |
| | Rate of residential beds |
| | Rate of residential beds in nursing homes |
| | Health care facilities for elderly |
| | Health care facilities for physically disabled |
| | Current public health expenditure for services provided directly (%) |
| | Index of territorial coverage of home care for the disabled |
| | Rate of family carers per population 65+ years old |
| | % people with university degree |
| | Activity rate, 15+ years old |

On the contrary, all the variables describing the material resources that highlight that the public system (e.g., the integrated home care rate or the rate of residential beds for the elderly in nursing homes) and the public's financial support (e.g., disability pensions or current public health expenditure for services provided directly) contributed to a reduction in the incidence of HP. Their contribution was supported by some personal and family characteristics: a high education level, a good activity rate, and families with few members.

The variables in Table 1 can be classified into three typologies:

- Burden of care;
- Provided services;
- Demographic and socio-economic characteristics of individuals and families.

The presence of these three categories supports the validity of the study's conceptual framework.

The first multivariate model (Table 2) shows the strongest increasing effect on poverty due to the increment of disability (2.089), followed by the attribution of the amount of public health expenditure rerouted through private structures (0.854).

**Table 2.** Linear regression model 1: Disability—Health system answer—Household Poverty.

| | Unstandardized Coefficients B | *p* |
|---|---|---|
| Standardized ADL Limitations Rate, 6+ years old, M + F | 2.089 | 0.047 |
| Current public health expenditure in total agreement for social benefits (%) | 0.854 | 0.013 |
| Average amount of total disability pensions | −0.005 | 0.014 |
| Health care facilities for elderly # | −0.009 | 0.058 |
| Rate of residential beds in nursing homes | −0.013 | 0.014 |
| Health care facilities for physically disabled | −0.031 | 0.028 |
| Integrated home care rate | −0.038 | 0.009 |
| Rate of residential beds | −0.039 | 0.012 |
| Integrated home care rate, 65+ years old | −0.060 | 0.028 |
| Index of territorial coverage of home care for the disabled | −0.148 | 0.025 |
| % Local Public Health Units with integrated home care service | −0.232 | 0.036 |
| Index of territorial coverage of home care for the elderly (65+) | −0.289 | 0.034 |
| Current public health expenditure for services provided directly (%) | −0.317 | 0.048 |

**Dependent variable:** Incidence of Household Poverty; adj. $R^2$ = 0.933; **Excluded variables:** % of elderly people (65+) treated in home care; Cases treated by integrated home care. # This variable is not statistically significant, but it remained in the model after all the tests.

In the opposite direction (reduction in poverty), covariates regarding the public health system's territorial services represented the most effective answer to ADL disability needs. The % of current public health expenditure for services provided directly (−0.317), the index of territorial coverage of home care for the elderly (−0.289), the % of local public health units with integrated home care services (−0.232), and the index of territorial coverage of home care for the disabled (−0.148) were the more efficacious contributors to the reduction in HP. Moreover, territorial facilities for disabled people (the rate of residential beds, health care facilities for the physically disabled, the integrated home care rate, and the rate of residential beds in nursing homes) supported this decrease. The direct financial contribution to the person (disability pensions) scarcely correlated to the reduction in HP (−0.005).

The second explorative multivariate model shows the role of education, employment, and household structure on familial poverty. The increase in HP was related to the growth in the rates of ADL disability (2.884), % of two and four family members (1.468 and 1.296), unemployment, and a low level of education (0.400 and 0.493) (Table 3).

**Table 3.** Linear regression model 2: Disability—Household structure—Activity—Household Poverty.

| | Unstandardized Coefficients B | Sig. |
|---|---|---|
| Standardized ADL Limitations Rate, 6+ years old, M + F | 2.884 | 0.046 |
| % 1 person families | 0.202 | 0.007 |
| % 2 persons families | 1.468 | 0.040 |
| % 4 persons families | 1.296 | 0.012 |
| % people with no education or primary level | 0.493 | 0.019 |
| Unemployment rate, 15+ years old | 0.400 | 0.006 |
| Rate of family carers per population 65+ years old | −0.001 | 0.008 |
| % people with university degree | −0.219 | 0.074 |
| Activity rate, 15+ years old | −0.818 | 0.007 |
| % 3 persons families | −1.344 | 0.024 |

**Dependent variable**: Incidence of Household Poverty; adj. $R^2$ = 0.973; **Excluded variables:** % 5 persons families; % 6+ persons families; Rate of healthy people, 65+ years old.

HP was counteracted by a specific family structure (% of 3 members families, −1.344), the activity rate (which considers people of 15+ years old, −0.818), and the presence of a high level of education (−0.219). The presence of family carers seems to have a very low (but statistically significant) effect in reducing HP (−0.001).

The first simple factor analysis synthesized the covariates in Table 2 in a factor that described how the system was able to counteract the effects of disability on HP (total explained variance = 84.9%).

The System's Answer Capability Index obtained in this way was correlated to ADL disability, and Scheme 1 illustrates the graphic evaluation of the relationship between ADL disability, system response, and the incidence of poverty at regional level (on a scale from 0 = lowest incidence of poverty and lowest response, to 10 = highest incidence of poverty and highest response).

The model outlines three regional groups:

- The "high response" group, mostly composed of northern regions, where the system's response was higher than average with respect to a low incidence of poverty;
- The "low response" group, composed of some central regions, where the system's response was lower than average and in line with the incidence of the poverty trend;
- The "low and out of the line response" group, composed of southern regions, where the system's response was low with respect to an incidence of poverty higher than the national average.

Figure 1 depicts how behaviour differs between northern and southern regions.

The second simple factor analysis synthesized the covariates in Table 3 in a factor describing how families can counteract the effects of disability on HP (total explained variance = 84.9%). The Household Support Index obtained in this way was correlated to ADL disability, and Scheme 2 shows the graphic evaluation of the relationship between disability, good health, familial structure and employment/unemployment, and the incidence of poverty at regional level.

The presence of two kinds of Italy emerges from the graph:

- A first Italy which was characterized by a combination of familial and occupational support, with a consequent low incidence of familial poverty: This situation characterized the central–northern regions, shown in the graph by the cluster which occupies the quadrant on the bottom left;
- A second Italy which was conversely marked by a high degree of difficulty on the part of families when responding to disabilities, with the incidence of poverty moving towards a high level. Southern Italy, however, showed a greater differentiation: Abruzzo, Molise, and Sardinia seemed, in fact, to have a slightly more favourable situation compared to the other southern regions.

Figure 2 depicts how behaviour differs between northern and southern regions.

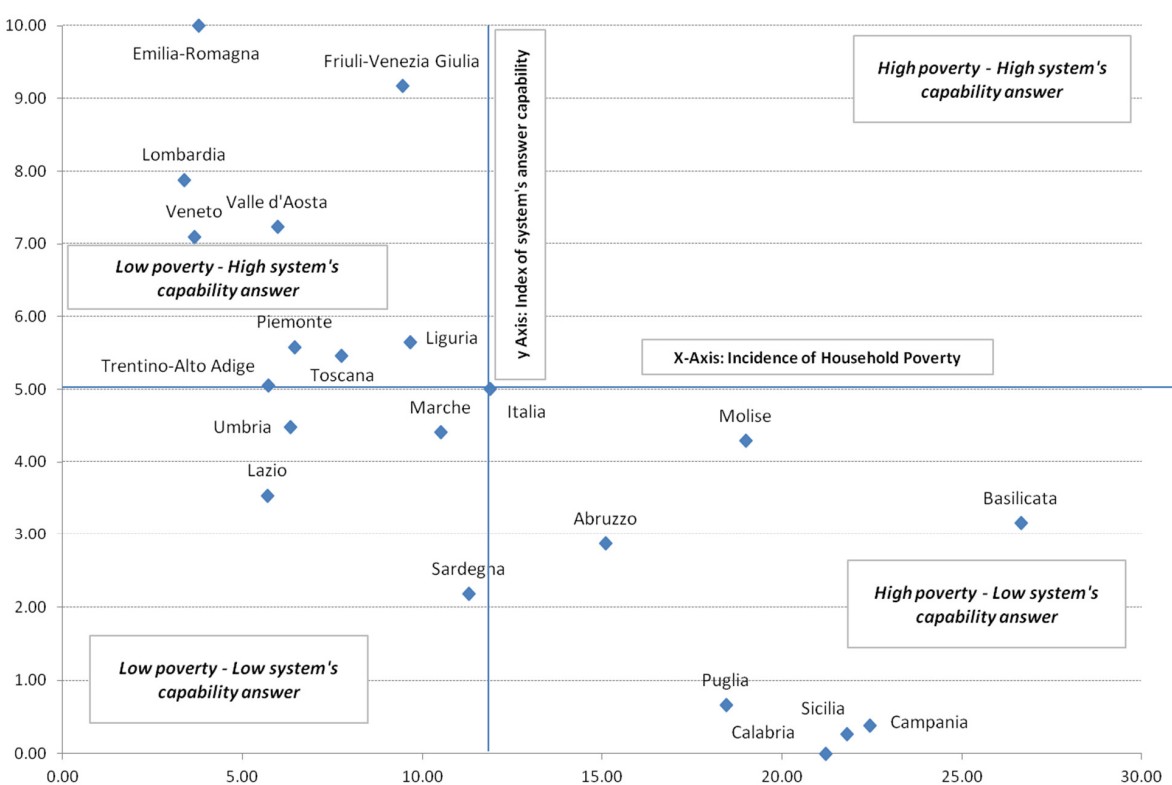

**Scheme 1.** Italian regions according to the combination of incidence of household poverty and index of system's answer capability: Index of system's answer capability ranging from 0 = lowest capability to 10 = highest capability.

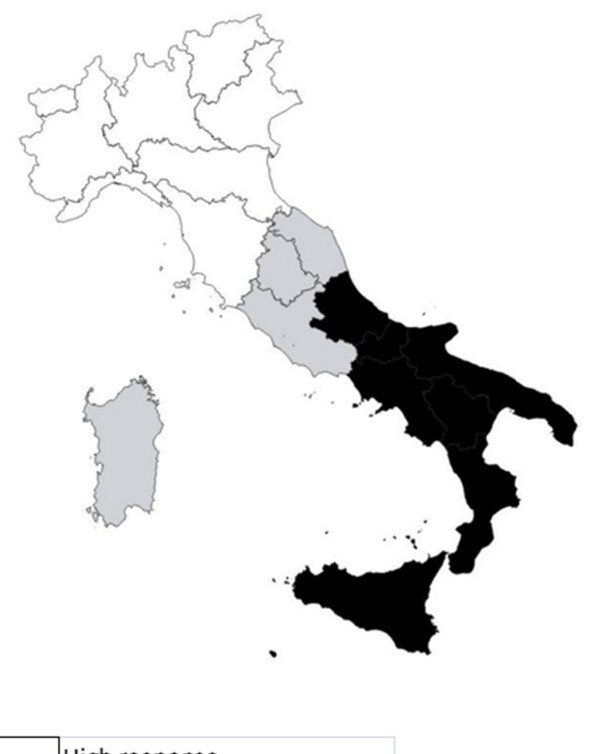

**Figure 1.** Italian regions' groups with respect to the system's answer capability.

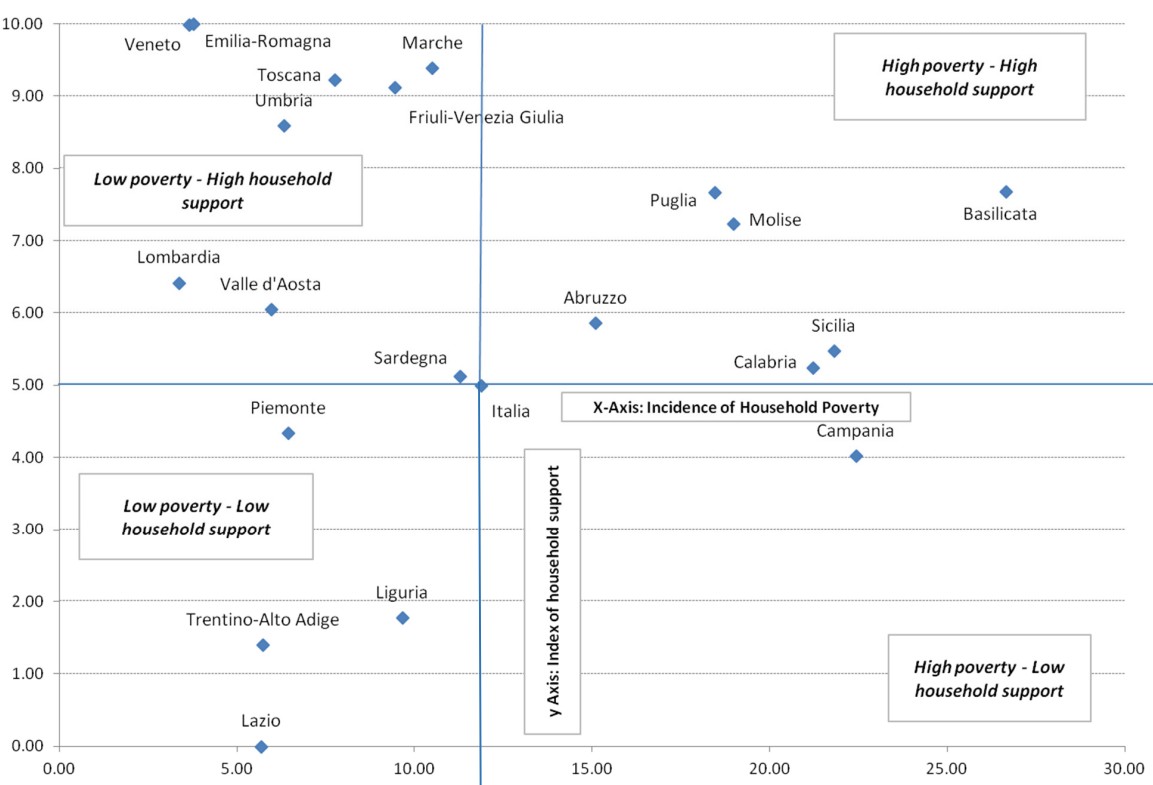

**Scheme 2.** Italian regions according to the combination of incidence of household poverty and index of household support: Index of household support ranging from 0 = lowest support to 10 = highest support.

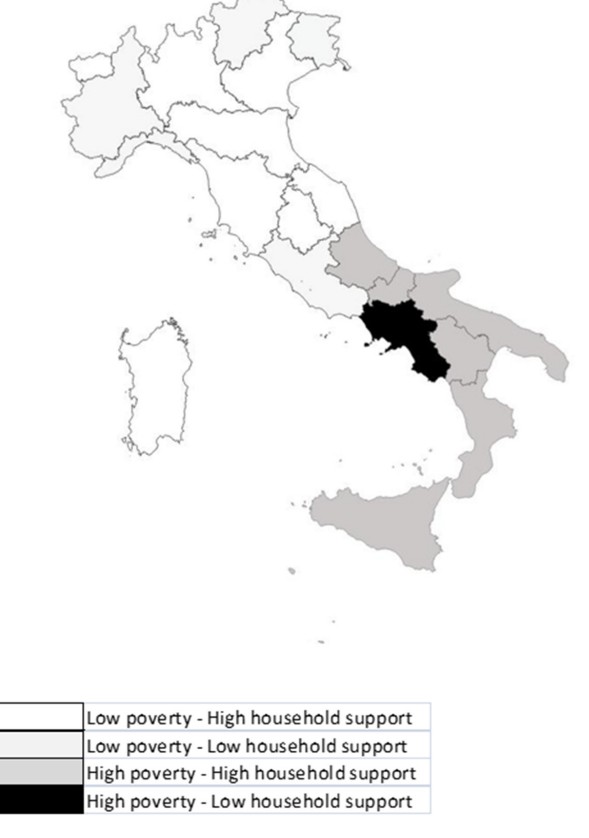

| | Low poverty - High household support |
|---|---|
| | Low poverty - Low household support |
| | High poverty - High household support |
| | High poverty - Low household support |

**Figure 2.** Italian regions' groups with respect to the household support.

## 4. Discussion

This study's results show the existence of a significant correlation between the presence of ADL disability and household poverty. In Italy, families that have at least one member with ADL limitations (often older people) are more easily exposed to the risk of socio-economic deprivation.

The analysis of variables involved in the correlation (Table 1) underlines how the increase in household poverty is influenced by two different typologies of variables: (a) "material resources unavailability", represented by the unemployment rate and low education rate; (b) the level of territorial care burden, mainly represented by the ADL Limitation rate and by the % of elderly people treated in home care.

Public health expenditure in the form of social benefits works as a proxy for care needs. In fact, it is supplied to cover social care needs [43]. In Italy, living in "a large family" (composed of at least four persons) does not make it possible to adequately overcome the risk of household poverty, probably because care needs in a large family may extend to more members (e.g., children or grandparents) [44,45].

In this framework, also confirmed by the linear regression models, the presence and the intensity level of specialized formal care services (home care and residential or semi-residential care) demonstrates a relevant difference in overcoming household poverty and the related risk of socio-economic deprivation, particularly when integrated with the household answer capability.

However, the picture of Italy surfacing from this study highlights large differences by region. The two linear regression models depict an Italy represented as a nation with two very distinct faces: Northern and Central Italy, where formal and informal networks are integrated and contribute to reducing the effects of SE deprivation on families; and the South, where such supporting elements reach a lower level of real efficacy. The availability of public services on the territory is the more relevant contender of SE deprivation. Indeed, the cost of services, if sustained by general taxes and/or by limited forms of co-payment, significantly reduces the economic possibilities of the single individual or of the familial nucleus [5,43,46]. More specifically, regional distributions show how regions with a higher system answer capability invest more in LTC in terms of resources and services than others [47,48]. Their regional regulation assessment specialized in LTC is higher than in other regions [22], regardless of the chosen regional strategy and the involved stakeholder (e.g., public, private, and NGOs).

For example, regions such as Lombardy, Veneto (characterized by a private health care strategy), Piedmont, or Emilia-Romagna (with a public health care strategy) benefit from specific-oriented strategies with efficient results from the used resources.

On the contrary, Puglia is an example of how an oriented strategy (public in this case) cannot be enough to overcome the incidence of HP when the regional system presents a low answer capability to the care burden.

A well-working operative strategy seems to be an integrated one, including formal and informal care, as shown by the multivariate models. An oriented strategy should carefully consider how this aspect is to be implemented in order to optimize families' economic and practical efforts [49].

The study results recall the attention of international policymakers on the effects of the LTC strategies adopted for the sustainability of LTC systems and the socio-economic condition of the population [7,50,51]. In particular, in the countries where informal care is widely used, such as Mediterranean countries and Baltic countries, the families must be protected and supported by the collateral risk of socio-economic deprivation.

In the last years, European institutions and national governments have agreed to promote a sustainable strategy for LTC based on the valorization of the home care and the integration actions between formal and informal care [52]. Additionally, the study put attention on how the existing different local LTC systems could produce relevant differences for families. The multi-level governance of a multidimensional issue, such as LTC, needs a very strong collaboration and agreement between national and local institutions. The many

difficulties and interruptions met by the implementation of the regional LTC system in Spain confirm this assumption [53]. Moreover, the recent health crisis due to the COVID-19 pandemic confirmed the central role of informal caregiving for the European LTC [54], underlining the relevance of flexible LTC and health protection policies [55]. In 2020, the German government offered an explicative example of flexibility. In contrast with the ongoing LTC strategy oriented towards community care, the government rapidly adopted policies supporting informal care, achieving a good response in terms of support of LTC needs in the health emergency context [56]. Finally, a recent study underlined how the adoption of living standards or socio-economic evaluation tools support the better distribution of the available resources [57].

## 5. Limitations

There are some limitations of this study: The chosen coverage strategy has a strong impact on family deprivation. However, some of the details in smaller areas were lost, due to the presence of clusters of individuals with specific situations and needs, and which a study at regional level cannot describe. While this study can suggest general indications on how to address the necessary actions, an individual study at a local level could propose more accurate evaluations on smaller areas and promote the identification of individual indicators to support innovative policies.

## 6. Conclusions

The study has succeeded in verifying the existence of a statistically significant relationship between ADL disability and increasing poverty for families. In Italy, ADL disability works as a risk factor to families' socio-economic deprivation, particularly if the system's answer is inadequate. The integration of a high system answer and strong household support can help to counteract poverty, but the system answer seems to make a difference. Whereas the Italian population is one of the oldest in Europe, with high care needs that will increase in coming years, policymakers should give due attention to the availability of services and resources to promote innovative policy to counteract the risk of socio-economic deprivation in families, especially in Mediterranean countries with a familial care regime. Finally, a well-oriented strategy (including formal and informal care) can help to improve the allocation of the resources, supporting at the macro level the sustainability of the LTC system and, at the micro level, helping the families against the risk of socio-economic deprivation.

**Author Contributions:** Both the authors are equally contributors for every single aspects of the article (research question, study design, methodology and analysis, text writing, references choice, revision, etc.). All authors have read and agreed to the published version of the manuscript.

**Funding:** The study is being supported by the Marie Curie European Fellowship Grant. Horizon 2020 MSCA-IF-2019 Grant Agreement No. 888102. INRCA—Family International Monitor Project.

**Institutional Review Board Statement:** Not applicable.

**Informed Consent Statement:** Not applicable. The study used only aggregated data at regional/ national level routinely collected by the Italian National Statistics Office from regional administrative archives. No individual data was used in the study and no person was individually contacted for collecting information.

**Data Availability Statement:** Data supporting reported results can be found at: https://www.istat.it/it/archivio/14562 (accessed on 9 November 2021) and may be freely accessed.

**Conflicts of Interest:** The authors declare no conflict of interest. The funders had no role in the design of the study; in the collection, analyses, or interpretation of data; in the writing of the manuscript, or in the decision to publish the results.

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
