# Peer review of "Disability in Older People and Socio-Economic Deprivation in Italy: Effects on the Care Burden and System Resources"

_sustainability, doi:10.3390/su14010205_

Round 1

Reviewer 1 Report

By the authors own admission, this paper offers little new information and is presented in a very skewed and conversational manor that lacks the objectivity of a scientific paper. If the authors goals is to provide objective evidence then their paper should read as equally objective in the presentation of that evidence. Other journalistic sources can be used to editorialize the results outside of the scientific community.

 Avoid using acronyms in the abstract without defining them 1st.

Similarly, jargon is used throughout the introduction and paper that is not defined or explained.

This is not a popular press or journalistic publication. Authors include references to “recent” events and “the crisis” these may be clear to the readership now but require better explanation in a scientific journal that may be referenced in the future.

Relative terms like good (line 24), particularly (line 74), better (line 330) comments like “common sense” (line 82), policy makers “have to” (line 369) feel editorial in a scientific manuscript. Consider alternatives.

Drastic (line 151) is a sensationalized phrase, please use an alternative.

The authors judiciously use the phrases high, very high, low and very low throughout the manuscript. In most instances it is not clear what these are relative to and what constitutes each. There are subjective terms and are used in excess without context. These should be avoided or clarified.

Watch spelling (e.g carers line 236)

The font is variable in Graph 1 & 2 this should be corrected or explained.

Remove single from line 376.

Author Response

See Attached document

Reviewer 2 Report

The authors provide an interesting overview and insight of how the National Health System in Italy address disability, indicating strengths and weaknesses of the overall system and comparing the different regional capabilities.

As a unique minor point, I would include a definition for the poverty and absolute poverty conditions.

Author Response

Thanks for your comment and positive feedback. We introduce the definition of absolute poverty, underlining how literature for European countries, prefer used the  relative poverty to measure the risk of people's impoverishment in European countries. We hope this help to joint your request to our approach of research.

Reviewer 3 Report

I read the manuscript ‘Disability in older people and socio-economic deprivation in Italy: Effects on the care burden and system resources ‘ with interest. I recommend it to be published in its present form as it meets the criteria of a scientific paper and discusses a crucial and up-to-date topic for ageing societies.

A really minor thing is that limitations should make a separate section to show this element better: 5 Limitations and conclusions re-numbered as 6.

Author Response

Many thanks for your positive feedback, the revision requested has been done.